# Physics-based modeling provides predictive understanding of selectively promiscuous substrate binding by Hsp70 chaperones

Erik B. Nordquist[1], Charles A. English[2,3], Eugenia M. Clerico[2], Woody Sherman[1,4], Lila M. Gierasch[1,2]*, Jianhan Chen[1,2]*

1 Department of Chemistry, University of Massachusetts Amherst, Amherst, Massachusetts, United States of America, 2 Department of Biochemistry and Molecular Biology, University of Massachusetts Amherst, Amherst, Massachusetts, United States of America, 3 Impact, New York City, New York, United States of America, 4 Roivant Sciences, Boston, Massachusetts, United States of America

* gierasch@biochem.umass.edu (LMG); jianhanc@umass.edu (JC)

**Data Availability Statement:** All relevant data are within the manuscript and its Supporting Information files.

## Abstract

To help cells cope with protein misfolding and aggregation, Hsp70 molecular chaperones selectively bind a variety of sequences ("selective promiscuity"). Statistical analyses from substrate-derived peptide arrays reveal that DnaK, the *E. coli* Hsp70, binds to sequences containing three to five branched hydrophobic residues, although otherwise the specific amino acids can vary considerably. Several high-resolution structures of the substrate -binding domain (SBD) of DnaK bound to peptides reveal a highly conserved configuration of the bound substrate and further suggest that the substrate-binding cleft consists of five largely independent sites for interaction with five consecutive substrate residues. Importantly, both substrate backbone orientations (N- to C- and C- to N-) allow essentially the same backbone hydrogen-bonding and side-chain interactions with the chaperone. In order to rationalize these observations, we performed atomistic molecular dynamics simulations to sample the interactions of all 20 amino acid side chains in each of the five sites of the chaperone in the context of the conserved substrate backbone configurations. The resulting interaction energetics provide the basis set for deriving a predictive model that we call Paladin (Physics-based model of DnaK-Substrate Binding). Trained using available peptide array data, Paladin can distinguish binders and nonbinders of DnaK with accuracy comparable to existing predictors and further predicts the detailed configuration of the bound sequence. Tested using existing DnaK-peptide structures, Paladin correctly predicted the binding register in 10 out of 13 substrate sequences that bind in the N- to C- orientation, and the binding orientation in 16 out of 22 sequences. The physical basis of the Paladin model provides insight into the origins of how Hsp70s bind substrates with a balance of selectivity and promiscuity. The approach described here can be extended to other Hsp70s where extensive peptide array data is not available.

**Funding:** This work was supported by the National Institutes of Health through R01 GM114300 (to JC) and R35 GM118161 (to LMG). EBN was also supported by National Research Service Awards T32 GM008515 and T32 GM139789 from the National Institutes of Health. The funders had no role in study design, data collection and analysis, decision to publish, or preparation of the manuscript.

**Competing interests:** The authors have declared that no competing interests exist.

## Author summary

Molecular chaperones are proteins that help prevent misfolding and aggregation of their substrates. This is a complex task, as the cell is very crowded, and the chaperone must efficiently bind to misfolded regions of the client protein while avoiding well-folded proteins. An additional confounding detail is that proteins can enter the binding cleft of some Hsp70s in two orientations, a fact often unaccounted for by existing sequence-based models.

Here, we developed a model to describe how client proteins bind to DnaK, the *E. coli* Hsp70, using physics-based molecular dynamics simulations to quantify the interactions between a variety of peptide substrates and key sites on DnaK. The resulting model, which we call Paladin, provides a physical basis to understand how DnaK binds to specific peptides. Given a sequence, Paladin can predict the precise residues that bind at a specific site on DnaK and can further explain challenging features like the binding orientation, which are typically not predicted by sequence-only models. The Paladin model could be used to design 'super-binder' therapeutic peptides to inhibit chaperones like DnaK in *E. coli*.

## Introduction

To maintain a healthy proteome, the cell relies on an extensive network of molecular chaperones and degradation enzymes called the protein homeostasis (or proteostasis) network [1–3]. Central to these quality control networks are the highly conserved 70-kDa heat shock chaperones (Hsp70s) [4–7]. Hsp70s work with their co-chaperones––J-proteins and nucleotide-exchange factors (NEFs)–to facilitate a range of functions, including promoting protein nascent chain folding, inhibiting aggregation, stabilizing unfolded states for translocation across membranes, working with downstream chaperones and disaggregases, and more [6,7]. A key aspect of Hsp70s' functions is binding hydrophobic and likely aggregation-prone regions of their client proteins, which protects them from aggregation and promotes productive folding (7). Substrate binding to the Hsp70's C-terminal substrate-binding domain (SBD) is modulated by the binding and hydrolysis of ATP at the N-terminal nucleotide-binding domain (NBD) (Fig 1). When ATP is bound, the SBD is docked onto the NBD, and the α-helical lid of the SBD disassociates from the "β-sandwich" sub-domain of the SBD (βSBD in Fig 1) [4,6,8]. Substrate binding to the βSBD in the ATP-bound state stimulates ATP hydrolysis by the NBD. Upon ATP hydrolysis, the ADP-bound NBD and the SBD undock, and the α-helical lid of the SBD associates with the βSBD, promoting the stable binding of the substrate (slow on/off rates) [9]. Hsp70s bind, hold, and release their client proteins by cycling between these two allosteric endpoints.

Although Hsp70s must bind diverse client peptides with a high level of promiscuity to support a multitude of cellular functions [6], they do not bind to all proteins in the cell. That is, Hsp70 binding is "selectively promiscuous". How does the nature of substrate binding enable Hsp70s to carry out surveillance of the proteome? What are the molecular bases of Hsp70s selectivity and affinity for a sequence in a client protein? Biochemical data have shown that while Hsp70s often bind to short, 7-residue segments rich in hydrophobic residues flanked by positively charged residues [13–17], the bound sequences show high variability. Structural details of substrate binding have been provided by experimental structures of the DnaK (the well-studied *E. coli* Hsp70) SBD bound to model peptides [10,18–20] (Fig 2A). All high-resolution structures of peptide-bound SBDs show the substrate bound in a hydrophobic cleft in the SBD and in an extended conformation stabilized by several backbone hydrogen bonds with

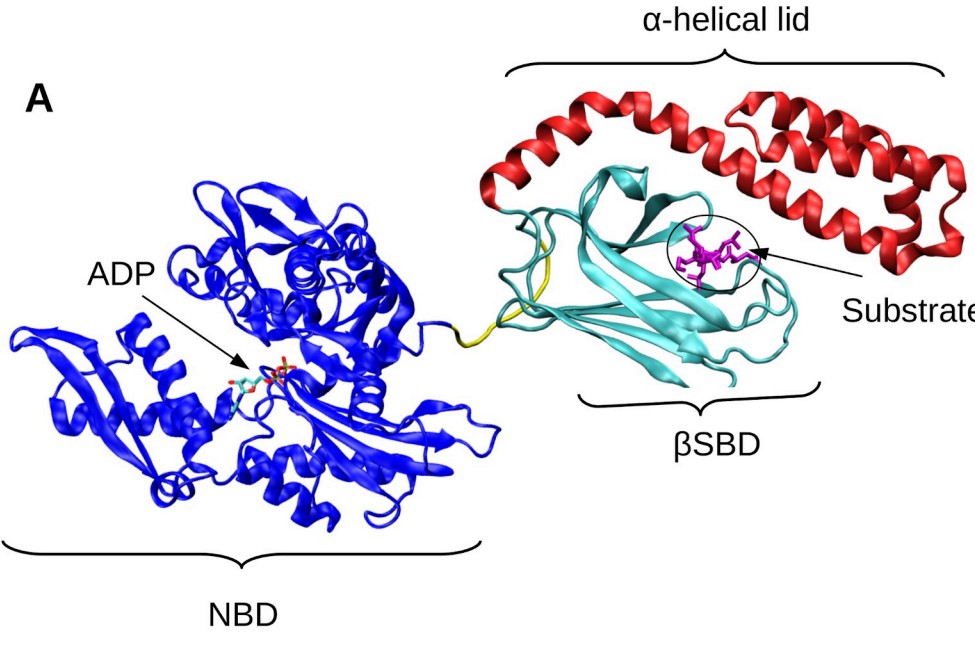

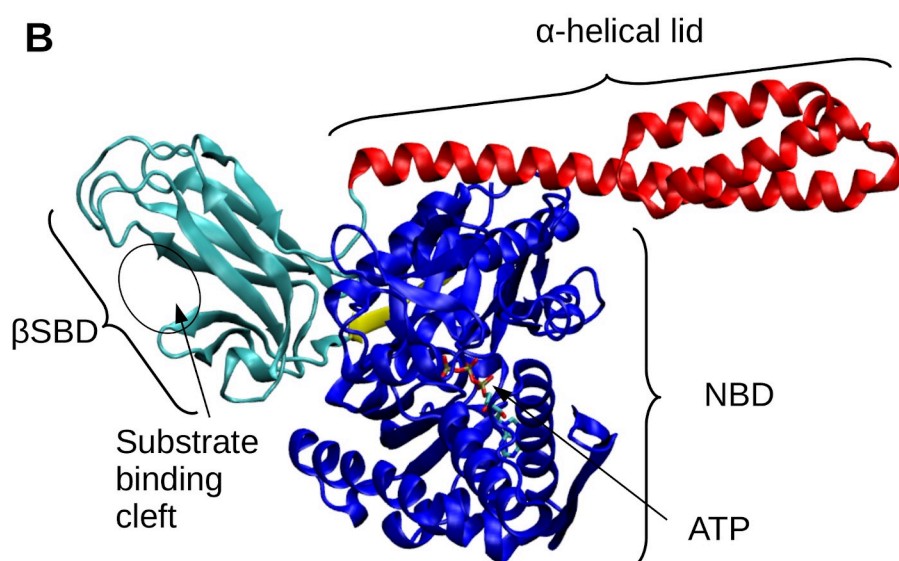

**Fig 1. Structural overview of DnaK in substrate-bound and -unbound states.** (A) Structure of DnaK in an unassociated (ADP-bound, substrate bound) conformation (PDB ID: 2KHO [9]). The nucleotide-binding domain (NBD) is shown in blue. The substrate-binding domain (SBD) has two subdomains: the α-helical lid is shown in red, and the βSBD is shown in teal. The substrate from the canonical NR peptide (PDB ID: 1DKZ [10]) is overlaid in the binding cleft of the βSBD in magenta. (B) Structure of DnaK in an associated (ATP-bound, no substrate) conformation (PDB ID: 4B9Q [11]). The ATP molecule is shown based on PDB ID: 3AY9 [12].

the SBD. Only five 'core' residues directly interact with the SBD using five sub-sites dubbed -2, -1, 0, +1, and +2 [10]. The central binding site in the chaperone (site 0) is the most stringent, as it consists of a pocket that is frequently occupied by a substrate L that is completely buried within the site.

The data obtained from peptide array studies have been used for training and testing data to build models that predict DnaK binding sites. Rüdiger *et al.* developed an algorithm based

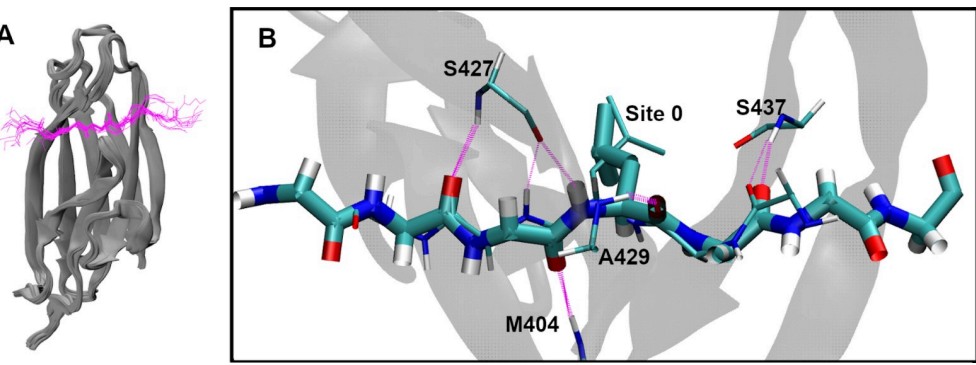

**Fig 2. Conserved βSBD-substrate binding conformation.** (A) Overlay of 18 bound substrate structures (S1 Table), aligned using the backbone atoms of the βSBD. The SBDs are show in gray cartoons, and the substrates in magenta backbone traces. Only the forward orientation substrates are included. (B) Overlay of two forward and reverse orientation peptide substrates (forward: NRLLLTG, PDB: 1DKZ [10]; reverse: NRLILTG, PDB: 4EZY [19]; underlines indicate residues at Site 0). The forward orientation is shown in thick bonds and the reverse in thin bonds. The backbone hydrogen bonds between βSBD and the substrate are shown in magenta dashed lines. Note that backbone hydrogen bonds involving M404, S427 and A429 completely overlap in two orientations. Side chains are shown only for the central L at Site 0. For more viewing angles of panel B, see S1 Movie.

on peptide array scanning of DnaK binding of 13-mer peptides derived from 37 proteins [15] where they defined a five-residue DnaK-binding core and four-residue flanking sequences on either side. From this information, they determined relative binding free energies from the frequency of each residue in these regions. However, this model does not provide site-specific binding mode information or any other structure-specific insights. Subsequently, van Durme *et al.* built a DnaK-binding prediction model from their own peptide array data [16]. To predict site-specificity, these researchers used the FoldX force field to find the optimal position for heptameric sequence binding in the forward orientation (N- to C-), as observed for NRLLLTG binding to SBD [10]. The resulting site-specific model, which is called Limbo, combines sequence- and structure-based binding energies into a position-specific scoring matrix (PSSM) of interaction terms. A position-independent matrix of FoldX-derived terms, Chaper-ISM, improves Limbo's ability to predict binding hotspots in peptide sequences but lacks any information about the positions occupied in the SBD or the orientation of the bound peptide [21]. Importantly, as an empirical energy function designed for modeling structured proteins and specific protein interactions [22], FoldX is limited in its ability to describe the inherent flexibility and dynamics present in DnaK-substrate interactions. Last but not least, none of the existing models of DnaK-peptide binding has yet considered the C- to N- (reverse) orientation of the bound substrate, which has been observed in high resolution structures [19,23] and seems to be energetically competitive with the N- to C- orientation [23–25]. At present, substrate binding in the C- to N- orientation has only been considered in an algorithm that predicts substrate binding to endoplasmic reticulum-localized Hsp70, BiP [26].

In the current work, we sought a deeper understanding of the nature of substrate binding to DnaK using physics-based simulations to develop an improved prediction method that not only recapitulates the sequence preference of DnaK binding but also provides the molecular detail of the bound state including site specificity and binding orientation. Existing structures of DnaK-substrate complexes suggest that the substrate backbone adopts a highly conserved binding conformation in either orientation to allow sampling of alternative side-chain interactions with largely independent surfaces on the chaperone. As such, we deployed mean field theory [27] and atomistic molecular dynamics simulations [26,28] to sample the interactions of a flexible substrate side chain with the receptor Hsp70 and quantify the interaction

energetics. This allows us to calculate a physics-based PSSM of DnaK-substrate interactions that requires minimal training based on experimental data. Such a model should also be more transferable and extendable [26], such as for describing binding in both orientations and for modeling the substrate specificity of other Hsp70s without extensive experimental data. Using the physics-based PSSM, we derive a simple linear model combining information from physics-based interaction energy terms and experimental peptide array data. The accuracy of this method, called Paladin (Physics-based model of DnaK-Substrate Binding), was evaluated by examining its ability to recapitulate the existing peptide array data as well as the binding registry and orientation of substrate peptides with available structures.

## Results

### Substrates bind to βSBD in a highly conserved conformation

To better understand the structural basis of substrate binding to DnaK we started by analyzing all 18 existing structures of the DnaK βSBD bound in N- to C- ("forward") orientation (S1 Table) [10]. For this, we overlaid these structures by aligning the backbone atoms of the βSBD (residues 393–502). Whereas the βSBD is able to undergo large conformational changes during the functional cycle of DnaK [29], the backbone is highly conserved in the substrate bound state and appears to undergo minimal rearrangement to accommodate different substrates [10,18–20] (Figs 2A and S1A). Based on the limited number of structures available, the substrates appear to bind in a consistent pose, although for simplicity we focus on substrates that don't contain P (Figs 1A and S1B). Furthermore, the average RMSD of the βSBD backbone atoms in all structures is less than 1.5 Å (S1A Fig). Importantly, this suggests is that one does not need to consider significant βSBD conformational changes in deriving a predictive model of βSBD-substrate interaction.

### Substrate side chains interact with independent, conserved DnaK sites in both orientations

The highly conserved conformation of the backbone of the SBD-substrate complex has important implications on how the side chains interact with DnaK. The bound peptides' extended backbone projects the side chains of the five interior residues in alternating directions, which interact with a set of well-conserved sites on the surface of the βSBD (Fig 3). Residues flanking the central five adopt a much more diverse structural ensemble (Fig 2A), where the side chains are largely solvent exposed with no well-defined binding pocket on the adjacent DnaK surface. Zhu *et al*. proposed that these same five sites play the biggest role in substrate binding to DnaK [10]. Beyond these five sites, the substrates' backbones are highly dynamic and the side chains are largely solvated, having minimal contacts with the βSBD. Strikingly, an overlay of peptides bound in both N- to C- and C- to N- orientations (Fig 2B and S1 Movie) reveals that the backbone conformations remain conserved and all five hydrogen bonding interactions between the substrate backbone and βSBD are formed in both orientations (thick and thin magenta dashed lines). Importantly, the highly conserved backbone conformation projects the side chains into identical binding pockets regardless of the orientation. For example, Fig 2B and S1 Movie illustrate that the slight shift of Cβ positions does not affect the placement of the central L into site 0. As such, the same set of well-defined binding sites likely dictates DnaK-substrate interactions in both orientations, and thus the preferred orientation of substrate binding likely arises from the energetic balance of side-chain interactions but not backbone interactions themselves.

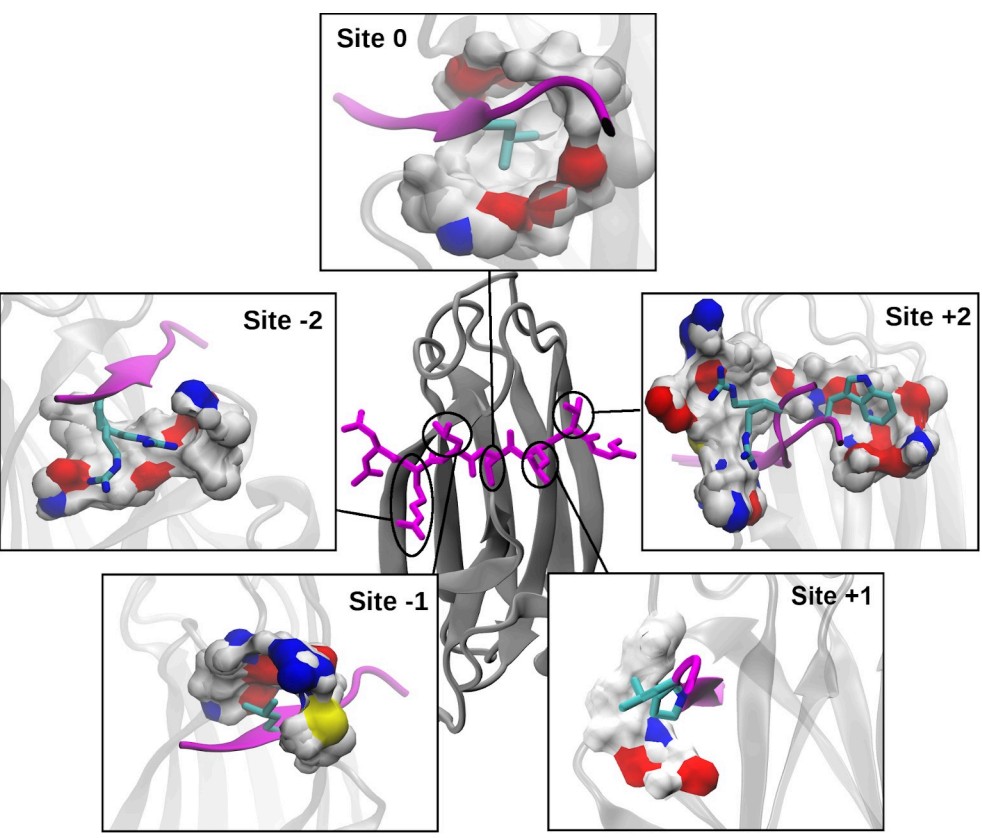

**Fig 3. Substrate-interacting sites of DnaK.** The center image shows a NRLLLTG substrate (magenta sticks) bound to βSBD (gray cartoon) (PDB: 1DKX). Each zoom-in shows the surface of one of the five binding sites on DnaK, colored by atom type: O (red), N (blue), C/H (white), S (yellow). The side-chain conformations shown were selected to clearly illustrate the extent of each site. These additional side-chain and substrate backbone conformations are taken from: (Site -2) 4JWD, (Site -1) 1DKX, (Site 0) 1DKX (for more viewing angles, see S2 Movie), (Site +1) 1DKZ, 4F00 and (Site +2) 4JWI, 4EZN, 4EZO. The binding site surface was generated using DnaK residues within 5 Å of the substrate side chain(s) at each site, with some atoms excluded for clarity.

## Binding site geometry and surface properties determine the DnaK-substrate selectivity

Analysis of the geometric and surface properties of the five sites on DnaK provides a qualitative understanding of the physical basis of promiscuous selectivity (Fig 3). Note that the central site 0 is the most sterically restrictive and hydrophobic pocket, which is known to contribute the most to selectivity [15,16] (S2 Movie). Its walls are composed of I401, F426, V436, I472, and V474, which allows hydrophobic residues to bury deep within the βSBD. From the available structures, it appears the central pocket mainly accommodates branched hydrophobic residues (S1 Table) as well as cyclohexylalanine (denoted Z in S1 Table) [15,16]. Indeed, Limbo predicts that primarily I, L, and M are favored at site 0, and V, T, and R to a lesser extent [16]. It is difficult to discern any site-specific information from the model from Rüdiger, *et al.*; it predicts that within the central 5-residue core region, L, I, V, F, Y, and R are preferred [15]. The central pocket is considered sterically too restrictive for bulky residues like W or Y [15,16]. Interestingly, there is a possible channel for long-chain charged or polar residues like R to partially escape from within site 0 through an opening to the +1 side of the pocket flanked by two backbone carbonyls (Figs 3 and S2). Furthermore, on the -1 side there are additional carbonyl groups that can interact favorably with shorter chain polar residues. The presence of

these carbonyls on the flanks of the pocket suggests that considering the desolvation of the pocket as well as that of the side chain could be important to describing total binding affinity.

The outer sites are qualitatively different from site 0, having concave patches along the surface of DnaK with which substrate side chains can interact. Site -1 is comprised of M404, Thr428, A429, and R467. As shown in Fig 3, it is the next most solvent shielded and likely prefers more hydrophobic residues such as L, P, and M, as well as aromatic residues such as Y or F. Indeed, both Rüdiger *et al.* and Limbo predict that, besides branched hydrophobic residues, larger aromatic F and Y are preferred at site -1 [15,16]. Because there is an easily accessible basic group (R) and a polar group (T) in addition to the non-polar groups, site -1 is a potential binding site for polar or charged residues (Fig 3). While Rüdiger *et al.* doesn't provide any indication that polar residues are preferred at any of the central five sites, Limbo suggests that E/Q are slightly preferred, while the shorter-chain N/D are strongly disfavored [15,16]. Site +1 is more solvent exposed than site -1 and is comprised primarily of E402, T403, M404, and A429. It accommodates mainly L, P, and T (S1 Table), indicating the ability to bind both hydrophobic and hydrophilic residues. Limbo predicts that many residues can favorably interact: branched hydrophobics (L, I, V), aromatics (Y, F), M, P, and R [16]. This is the only site at which Limbo predicts P will favorably interact, in strong agreement with crystallographic evidence (S1 Table) [16,19]. The bridging M404 that covers site 0 is partially shared by Site +1 and Site -1 could provide a mechanism for subtle cooperativity between these sites in substrate binding (Figs 3 and S3). M404 is a part of the hydrophobic arch which covers site 0 while the peptide substrate is bound and which is known to play a key role in locking substrates in place during binding [30,31].

The outer two sites are both broad basins on the surface of the βSBD and are each comprised of many polar residues. Both outer sites have structures with R in distinct conformations (Fig 3). Conformational heterogeneity at these sites along with their open natures suggest significant side-chain dynamics and few specific interactions. Large available surface area and lack of steric hindrance make these outer sites favorable to bulky aromatics like W. At site -2, Limbo predicts that branched hydrophobic residues (I, L, V), aromatic residues (F, Y), along with K and R, are preferred. The preference of positively charged residues at the N-terminus is consistent with existing structures including that of the canonical NR- peptide (S1 Table) [10]). Site -2 is distinct from site +2 in than it displays some preference for branched hydrophobic residues in the available structures (S1 Table). In contrast, Limbo predicts site +2 to have indistinct preferences, except for a strong preference for W [16]. Site +2 is the only site Limbo predicts to be favorable for W [16], and the only site where W is supported in an experimental structure (S1 Table) [19].

## Construction of a new physics-based model of DnaK-substrate binding

We further examine if one could derive a better predictive model by quantifying the energetics of side-chain interactions with each binding site on DnaK. Our overall strategy is to use physics-based atomistic simulations to sample substrate side-chain interactions with DnaK and calculate a set of average interaction energies. These energetic terms are then used as a PSSM basis set to train the Paladin model based on existing peptide array data to quantify the physical origins of selective promiscuity of DnaK.

## Physics-based interaction energy terms

The key to deriving the Paladin model is the matrix of independent energy terms of size 20 (residue types) x 5 (binding sites). It disregards contributions of residues beyond the five central sites. Based on our above structural analysis, this should be a reasonable place to simplify the model while retaining the most critical interactions, despite previous evidence that

including the statistical preference for residues in a region beyond the central five can improve agreement with peptide array data [15]. For the energy terms, we first considered six major physical forces, including van der Waals (vdW) interactions, electrostatic interactions, backbone strain, desolvation, backbone conformational entropy, and side-chain conformational entropy. The backbone strain captures the energetic cost of potential perturbation to the conserved βSBD and substrate backbone conformation to accommodate a specific side chain. For this, we impose mean harmonic restraint potentials on all backbone heavy atoms, with force constants that are consistent with the observed fluctuations in the existing bound structures (see Methods). Since P would incur unusually high backbone strain in the β-like conformation, P in the substrate was not directly restrained but held in place by its neighbors. The backbone strain energy was then calculated as the average harmonic restraint violation energy during atomistic simulations (see Methods). All averaged interaction energies were calculated between the substrate side chain and the rest of the complex (including the substrate backbone). The free energies of desolvation were estimated directly from the percent solvent-accessible surface area (SASA) change of the side chain upon binding, based on experimental transfer free energies of amino acid side chain analogs. This is similar to how FoldX solvation terms are estimated [32] and is a well-established method for estimating nonpolar solvation free energy [33–35]. The desolvation terms were calculated separately for substrate side chains and the site (everything else). The desolvation free energy cost of the backbone was calculated using the transfer free energy of the N side-chain as an analog. The entropic cost of locking the substrate into the highly conserved backbone conformation (Fig 2) was estimated from the coil propensity of the pentapeptide sequence. The entropic cost of side-chain rotation restriction was estimated based on the Dunbrack rotamer statistics [36] (see Methods).

The energy terms for site 0 are summarized in Fig 4 and the other sites in S4 Fig. As expected, the van der Waals interaction energies are roughly correlated with the size of the side chain in the confined environment, and electrostatics play a much smaller role in the largely hydrophobic site 0 (Fig 4 and S2 Movie) but are more energetically important in the outer, solvent exposed sites (S4 Fig). Contrary to our expectations, the backbone strain term was not particularly large even for bulky residues like W or F at site 0, suggesting that the site is not as sterically-hindered as previously believed and that minimal backbone adjustment is required (Fig 4). The desolvation term for the substrate side chain appears to largely recapitulate the expectation that residues like L and I are preferred at site 0, owing to a combination of their ideal fit (large percent surface area change) and hydrophobicity (Fig 4). It also suggests that charged residues are unfavorable, and that R is particularly unfavorable at site 0 due to a high transfer free energy. In this score matrix, the most important features are present: a clear preference for L, I, V, and M at site 0, more ambiguity at sites -1 and -2 for residues including I, L, V, M, C, and F, an increasing preference for large aromatic residues toward the open outer sites, and positively charged K is favored at outer sites +1 and +2. R's overall score at these sites is still unfavorable due to the large transfer free energy and thus desolvation term, but it is possible that K at sites +1 and +2 captures the weak preference for positively charged residues at these sites. Surprisingly, the distribution of interaction energies for each residue is very similar across each site (S5 Fig). This might explain why the position-independent method adopted by ChaperISM is effective (19).

## A linear model for DnaK-substrate binding: Model parameterization

To derive the Paladin model trained using peptide array data, we chose to use a simple linear model of terms that require only a small number of parameters,

$$E(\text{five−mer}) = w_{cp}E_{cp}^{\text{five−mer}} + E_{reverse} + \sum_{sites} w_{site} \sum_{terms} w_{term}E_{site,term}^{res} \tag{1}$$

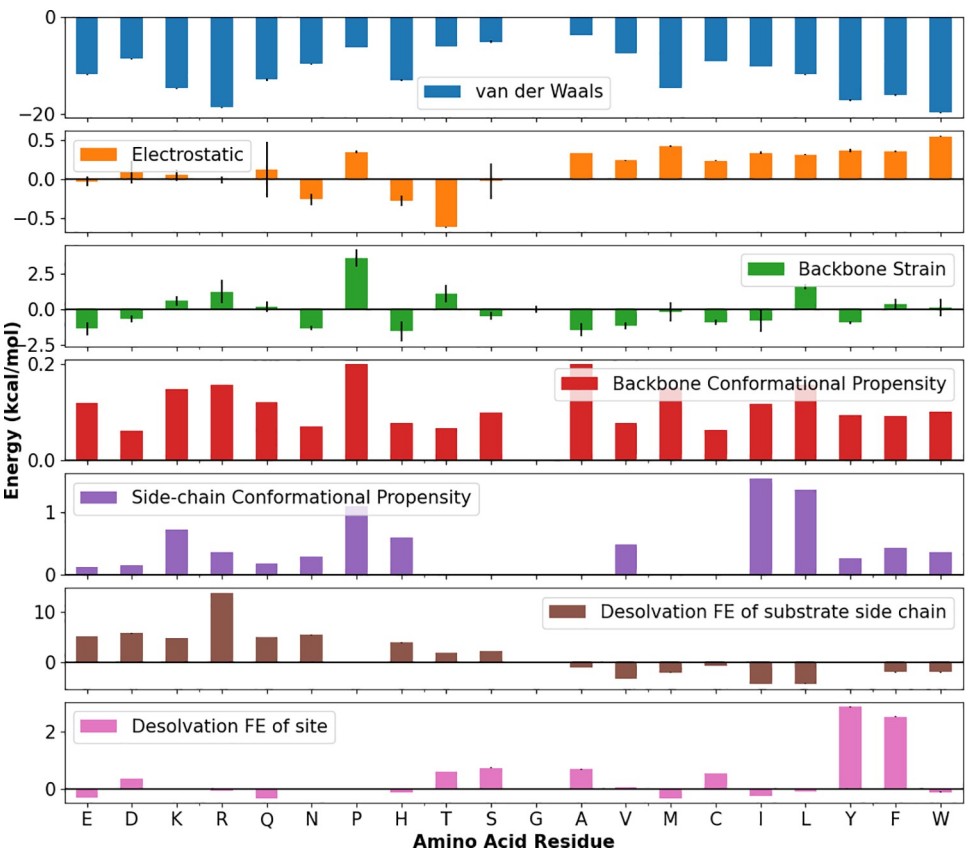

**Fig 4. Individual, unscaled physical interaction terms for all possible substrate amino acid side chains occupying the site 0 in the SBD.** Error bars are calculated as standard error of the mean. The residues are ordered by Wimley-White interfacial hydrophobicity scale to facilitate easy reading [37]. Error bars report standard error of the mean from the MD simulations.

where each $w_{term}$ is a weight to scale each term, and each $E^{res}_{site,term}$ is an energy term for a specific site and residue. $W_{cp}$, $E_{cp}$ are the weight and energy term corresponding to the mean coiled propensity for the whole five-mer, and $E_{reverse}$ corresponds to a small energy penalty if the peptide binds in the reverse orientation. The simple linear model with only 12 free parameters helps avoid over fitting the low-resolution peptide array data. Non-linear models, such as random forests or neural networks were also tested and they do not improve accuracy, and we expect the added complexity would make them less conveniently interpretable and less transferable. Importantly, a linear model reflects the physical nature of DnaK-substrate binding, that the specificity arises mainly from largely independent substrate side-chain interactions with conserved sites on the βSBD.

We used a Monte Carlo search to select weights that maximize the correlation between the resulting score and the peptide array data from Rüdiger *et al.* [15] (see Methods). Due to the semiquantitative nature of the assay, 13-mers had been grouped into four classes of peptides: strong-binders (high fluorescence), binders, neutral (ambiguous), and non-binders. We kept these classifications in order to train DnaK on the most important signals in the data as identified in the original study. We randomly selected 80% of that data to use as a training set; the remaining 20% was reserved to validate the final choice of weights (Table 1 and Fig 5). Note, during optimization we found that the weights of the backbone strain, and side chain and backbone conformational propensity terms had minimal positive effect on the correlation

**Table 1. Term and site weights and reverse orientation penalty of the Paladin model.**

| | | | |
|---|---|---|---|
| Van der Waals | 0.1 | Site -2 | 0.5 |
| Electrostatic | 0.6 | Site -1 | 0.5 |
| Backbone Strain | 0 | Site 0 | 1.0 |
| Backbone Conformational Propensity | 0 | Site +1 | 0.2 |
| Side-chain Conformational Propensity | 0 | Site +2 | 0.1 |
| Desolvation FE of Substrate side chain | 1.0 | | |
| Desolvation FE of βSBD site | 0.4 | Reverse penalty (kcal/mol) | 0 |

between resulting scores and the peptide array data (S6 Fig). While these three terms should capture some important physical properties of substrates like high flexibility or bulky side-chain strain, their effects on binding specificity are likely weak and the peptide array data may be too noisy to capture these effects. As such, the corresponding weights were set to zero in the final model (Table 1). Similarly, we found that the site weights (except site 0) didn't strongly influence the ability of the model to recapitulate the peptide array data (S6 Fig), which may also be attributed to limited site-specific information in the experimental data. The final site weights were empirically chosen to reflect existing understanding of the relative importance of each site to binding (see above), and these site weights are included in the combined score 5x20 matrix (S5 Fig).

The choice of weights for different energy terms also makes sense given the qualitative picture of binding described above. The terms which contribute the most at each site are the desolvation of the substrate side chain and the Van der Waals interaction energy. These terms reflect the importance of the hydrophobic effect and solvent entropy and non-specific contacts between ligand and protein in ligand-protein binding. Electrostatic interactions also contribute for charged residues, especially at some outer sites. The backbone strain and both conformational terms still likely contain relevant information about protein-ligand interactions, but the peptide array data may be too low resolution for our optimization method and linear model to include those terms. The site weights had relatively flat distribution in z-scores (S7 Fig), so they were selected to mimic the expected distribution described in the qualitative picture of binding above; that is, site 0 dominates, sites -2 and -1 contribute less than site 0 and

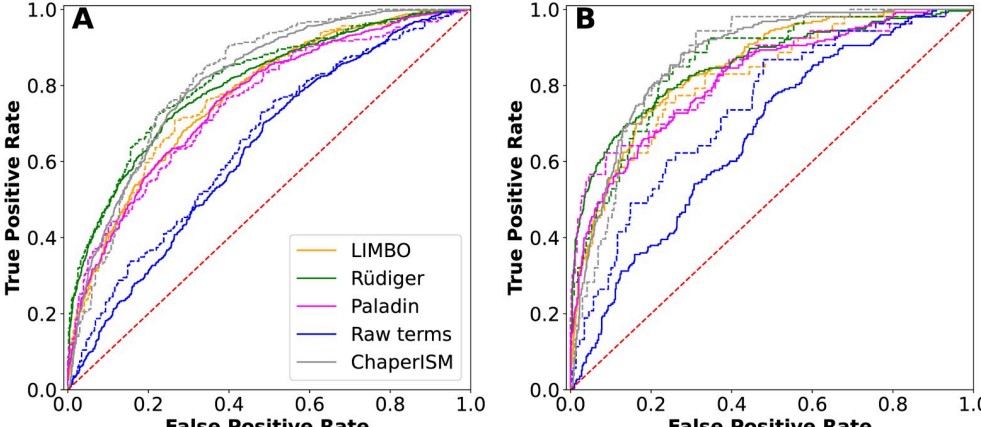

**Fig 5. Receiver operating characteristic curves of various predictors.** The raw score was calculated using the energy terms with uniform weights. (A) Discrimination of both strong binders and binders from nonbinders, and (B) Discrimination of strong binders from nonbinders. All solid lines were derived from the training set and dotted lines from the validation set. The red dashed diagonal line represents a random prediction.

about equally to each other, and sites +1 and +2 contribute minimally, with site +2 contributing the least. This choice of site weights is not unique and not necessarily optimal, but there was insufficient data to consider training them more carefully than this. We considered adding a purely empirical reverse penalty term to reverse-orientation scores to maximize the difference between predicted true forward and true reverse binding peptides and found that the penalty of 0 is sufficient (Fig 7, see binding orientation section for more details).

## Paladin can recapitulate the peptide array data

We used receiver operating characteristic (ROC) curves to examine how well the final Paladin model recapitulates the peptide array data, ROC curves provide a quantitative way to summarize the agreement between peptide arrays and predicted energy scores (an example predicted peptide array is given in S8 Fig). As summarized in Fig 5 and Table 2, the ROC curves and areas under those curves (AUC) are shown for all binders versus non-binders and strong binders versus non-binders (Fig 5A and 5B, respectively). Note that using the raw energy terms directly (i.e., Eq 1 with uniform weights) can reasonably discriminate both binders and strong binders from nonbinders, with (AUC) ranging from 0.63 to 0.74 (Table 1). This result demonstrates that these MD-derived energy terms indeed capture the true physics of DnaK-substrate interactions. The optimized Paladin model performs similarly to Limbo but appears slightly worse than the original Rüdiger, *et al.* algorithm or ChaperISM. For example, Paladin had a validation AUC of 0.83 in classifying strong binders from non-binders, where by comparison the Rüdiger, *et al.* [15] method and ChaperISM [21] had an AUC of 0.86 and 0.87, respectively. Re-weighting our energy terms to recapitulate this peptide array data comes with some important caveats. Since only 5 residues interact directly at well-defined sites on DnaK, each 13-mer peptide in the Rüdiger data set has 9 possible five-mers, creating significant ambiguity about which residues interacted at which site (registry). The binding affinities are only semi-quantitative and assigned to 4 categories. Additionally, the arrays contain no information about orientation, so we only scanned in the forward orientation, in line with the common expectation that most peptides bind in the forward orientation (16, 17). These inherent uncertainties seem to limit the ability of all predictive models in recapitulating the peptide array data.

Additionally, the peptide array dataset was constructed directly from protein sequences and therefore imbalanced, with far more nonbinders than binders (48% nonbinders versus 30% binders, with 12% classified as neutral, see Methods for details). Using an ROC curve to measure correlation in an imbalanced dataset can be misleading; a predictor which mostly predicts nonbinders can perform well, even if it frequently misclassifies true binders. Precision-Recall (PR) curves compensate for this problem by accounting for how the model predicts nonbinders as well. PR curves of the predictions of the same predictions, shown in S9 Fig with corresponding AUCs in S4 Table, reveal that in fact the original Rüdiger model has the most balanced description of the data, which makes sense since it was derived directly from the data

**Table 2. Area under the ROC curves (Fig 5).** The curves were based on the discrimination of all binders from non-binders, and of only strong binders from non-binders.

| Model | All binders | | Strong binders | |
|---|---|---|---|---|
| | Training | Validation | Training | Validation |
| Rüdiger, *et al.* [15] | 0.80 | 0.81 | 0.85 | 0.86 |
| Limbo [16] | 0.77 | 0.77 | 0.85 | 0.82 |
| ChaperISM [21] | 0.81 | 0.82 | 0.87 | 0.87 |
| Paladin | 0.76 | 0.75 | 0.82 | 0.83 |
| Raw Terms | 0.63 | 0.65 | 0.67 | 0.74 |

and should reflect a near maximum of the information one could derive from the data alone. On the other hand, the other models including Paladin perform comparably and the differences in ROC/PR curves are small. Importantly, apparent better fitting to the data do not necessarily reflect a superior model. Instead, by incorporating the physics of molecular interactions and using only a small number of free parameters (Table 1), the Paladin model is designed to be capable of predicting key molecular details of binding including backbone orientation and residue registry. This provides an opportunity to reveal the balance of various interactions from different sites in binding specificity and to model other Hsp70s without extensive peptide array data, only the relevant interactions matrix.

## Paladin can correctly predict and explain substrate binding registry

We examine the ability of Paladin to correctly predict substrate binding registry using existing experimental structures of complexes (S1 Table). Among these, there are 13 unique forward-bound substrate sequences in the PDB with more than five naturally-occurring (S1 Table). The results, summarized in Fig 5, show that the true binding registries have lowest Paladin scores in 10 of 13 cases, exceeding the performance models which outperform on the peptide array data (S10 Fig). Limbo has a relatively aggressive cutoff for binder scores, meaning almost none of these peptides classify as binders. Ignoring the cutoff, Limbo can clearly predict which seven-mer will bind in 9 out of 12 cases when considering only the forward orientation. ChaperISM on the other hand has a generous default cutoff and identifies nearly every seven-mer as a binder, and cannot discriminate between registers, because it was designed to scan for hotspots, which it does well (Fig 5). Interestingly, Paladin scores the most correct registers between -5 to -10 kcal/mol, consistent with the physical range of actual binding free energies. Many five-mers containing P, especially at site +1, are predicted correctly, even though the presence of P leads to significant perturbation to the conformation of substrate backbone and the binding site. In two cases, RPPRLPRPR and RRPRLPRPR, the correct five-mer (PRLPR in both cases) is lowest in energy by less than 0.2 kcal/mol (S2 Table). This five-mer only scored -3.3 kcal/mol, somewhat higher in energy than other binding pentamers due to the presence of R at site +2. The only forward-bound peptide Paladin predicts to bind reverse, PRPLPFP, also has an R at site -2, whereas the reverse pentamer (RPLPF bound C- to N-) puts the F at site -2. The fact that these sequences all have the same or similar central motif may overemphasize the stability of R at site -2, but it is possible as previously discussed that R's highly unfavorable transfer free energy and thus desolvation term may penalize it too much. A more accurate treatment of solvent effects could further improve the accuracy of Paladin. Our treatment of sidechains like R underestimate the solvation free energy, which could be mitigated by treating each sidechain as a whole unit rather than accounting for the drastically different polar head and hydrophobic tail. Additionally, the use of a SASA-based solvation model will systematically underestimate long-range effects arising from protein-water electrostatic interactions (see the Methods section for more discussion of these limitations).

The physics-based interaction terms and the linear model in Paladin provide an opportunity to further dissect the molecular basis of predicted substrate binding preference and registry. As an example, we examine the binding properties of NRLLLTG (PDB IDs: 1DKZ, 4EZW) and NRLMLTG (PDB ID: 4EZX) (Fig 6 and S2 Table). The fact that switching the middle residue from L to M results in a register shift of the peptide while maintaining its forward orientation can be explained by the benefit of -2.4 kcal/mol both for shifting L to site 0 instead of M, and for keeping the other L and M at sites -2 and -1 (forward orientation) (S2 Table). As a result, Paladin predicts that the five-mers LLLTG and LMLTG are predicted to bind most favorably. Such a quantitative prediction is also consistent with the previous notion that sites

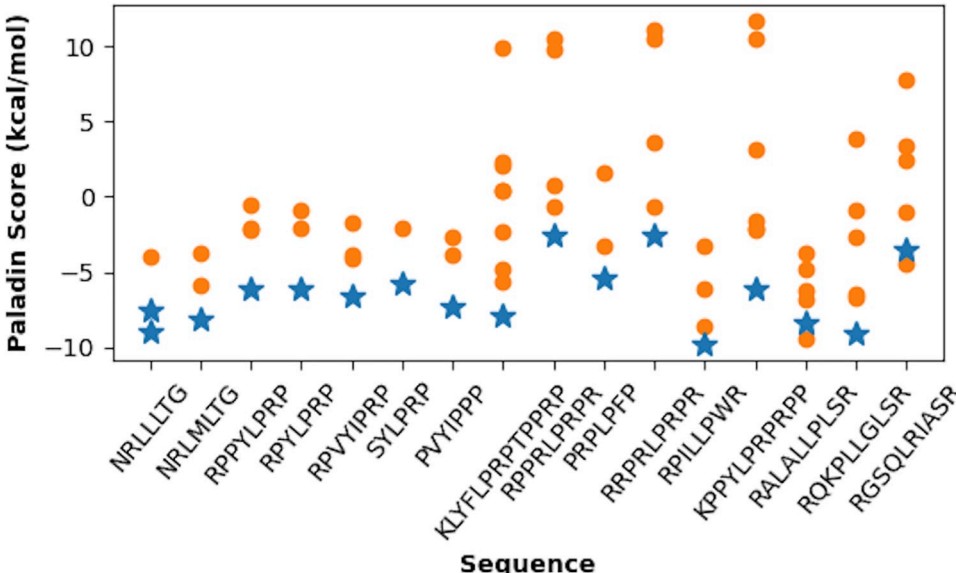

**Fig 6. Registry predictions for forward-binding substrates.** For each substrate, the Paladin scores are shown for all possible binding registries that allow all five sites on DnaK to be occupied. Both backbone orientations are considered (forward: blue; reverse: orange). The registries as observed in the PDB structures are shown with blue stars. Note that the NRLLLTG peptide has been crystallized in two registries (PDB: 1DKZ, 4EZW).

-2 and -1 have stronger preferences for hydrophobic residues than sites +1 or +2 [16]. Nonetheless, the observation that the other known binding five-mer NRLLLTG (PDB ID: 1DKZ) is not lower in energy than several of the reverse orientation predictions could reflect a limitation

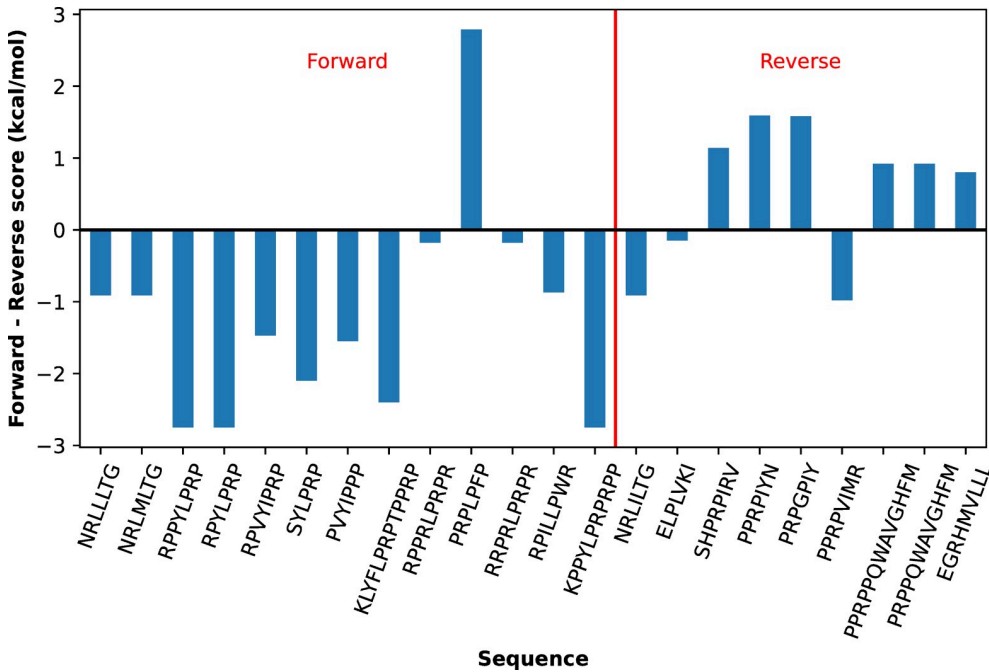

**Fig 7. Prediction of substrate binding orientation by Paladin.** The difference between the lowest forward and reverse scores will be negative when the for5tward orientation is favored, and vice versa. Substrates that actually bind in the forward orientation are labeled on the left of the red dividing line, and those that actually bind reverse are on the right.

in Paladin's ability to discriminate these similar sequences. But it is worth noting that NRLILTG (PDB ID: 4EZY) binds in the reverse orientation, suggesting that this basic motif presents a variety of potential binding registries and orientations. Interestingly, the observed binding registries share some prominent features: it places a branched hydrophobic residue at site 0 and a maximal number of other hydrophobic or aromatic residues at the sites -2 and -1. Correspondingly, the energetic terms which dominate Paladin's predictions are the van der Waals and desolvation FE for the substrate side chain. Paladin predicts that a binding sequence should maximize the hydrophobic surface area contacts primarily at sites -1 and -2 and minimize the hidden solvent entropic penalty captured in the desolvation term for both aromatic and hydrophobic residues at those sites.

## Paladin can discriminate forward and reverse binding orientation

Recent experimental studies have suggested that the reverse binding orientation is more prevalent for DnaK than was previously thought [24]. Even though only forward orientation was considered when training the Paladin model using the peptide array data, the observation that both orientations project the side chains towards the same set of sites on the βSBD (Fig 2B and S1 Movie) suggest that Paladin may be directly applied to predict the binding orientation and registry in both orientations. For this, we first examined if side-chain interactions for a substrate in the reverse pose would result in similar energy terms. We calculated the interaction energy terms for 10 representative residues at site 0 in the context of a reverse backbone conformation (taken from PDB ID: 4EZY), using the same atomistic simulation protocol (see Methods). As summarized in S11 Fig, the results confirm that all interaction energy terms are highly similar regardless of the substrate backbone orientation. As such, we considered only a single additional parameter, $E_{reverse}$, to capture the free energy difference between the forward and reverse backbone interactions, which is expected to be small. This value could be empirically selected to maximize the separation between the clusters of forward and reverse binders, but which has no real impact on this model's ability to resolve the binding orientation (Fig 7). We found that $E_{reverse}$ = 0 kcal/mol appears to be optimal with the current Paladin model.

In order to determine whether Paladin can discriminate both forward and reverse orientation, we marked the predicted orientation based on the forward and reverse five-mers with the lowest scores for each peptide in Table 1. As summarized in Fig 7, Paladin achieves ~73% accuracy, predicting the correct orientation for 16 out of 22 peptide substrates with structural data. Note that even though Paladin gives the correct sign for the free energy difference between two orientations for RPPRLPRPR and RRPRLPRPR, the values are significantly smaller than kT or 0.6 kcal/mol. As such, the binding orientation of these two peptides are considered ambiguous instead of correctly predicted. Limbo performs only slightly worse, with a success rate of 14 out of 21 cases where the peptide has at least 7 residues [16] (S12 Fig). This is impressive, given that Limbo was not developed considering reverse orientation, but likely owes to the underlying physical basis of Limbo's FoldX-derived PSSM. On the other hand, these success rates are superior to BiPPred, which predicts the correct orientation of the same dataset in 9 out of 22 cases, and only correctly predicts two in the reverse orientation [26] (S13 Fig). This likely owes to the fact that BiPPred was designed to recapitulate the binding interactions of a different Hsp70 chaperone.

One important caveat that should accompany the use of the reverse orientation structures is that they come from only a single study and often contain P residues, and so may not be representative of all reverse binding peptides. Additionally, we note that the above evaluation does not require Paladin to correctly predict the binding registry in both orientations. A closer inspection suggests that Paladin is more limited in predicting the binding registry of peptides

in reverse orientation (S14 Fig). Nonetheless, the experimentally observed registries mostly have Paladin scores only slightly worse than the lowest energy registry predicted. It is possible that the model could be improved by re-calculating an independent set of interaction energy terms for the reverse binding pose, but this does not address the limitation of the existing peptide array data used for modeling training, where no orientation information is available. Nonetheless, the apparent ability of Paladin to predict substrate binding orientation given how its basis set was derived suggests that Hsp70 has likely evolved to recognize or 'read' sequences in either orientation and the preferred binding orientation and registry is largely determined by the presence of side chains.

A challenging but important example of the sequence dependence of substrate binding orientation involves peptides NRL<u>L</u>LTG (two binding registers observed) and NR<u>LI</u>LTG. The first peptide binds in forward orientation, while the second one binds in reverse. It is not obvious why replacing the central L with I in NR<u>LI</u>LTG causes the peptide to flip orientation, as both L and I are hydrophobic residues similarly preferred at site 0. Unfortunately, Paladin is not able to recapitulate the orientation flip, which is both a sign of the limitations of the model and the difficulty of this case. It also highlights the limitation of Paladin that, despite previous evidence from peptide array data that L is more strongly preferred in site 0 than I, Paladin predicts that preference to be only -0.18 kcal/mol, about 5-fold lower what is expected based the frequency of L v. I at site 0 in crystal structures. For the true reverse orientation examples, it is difficult to make detailed analysis because Paladin gets the specific registry predictions wrong, even though it predicts another five-mer will bind in the reverse orientation (S13 Fig). In many of these cases the residue at site 0 is P, which neither Paladin nor Limbo nor the original Rüdiger model predicts will bind favorably at site 0.

To further test Paladin's predictive ability, we have scanned the Rüdiger data set in both directions, and selected the 100 peptides predicted to bind in the reverse orientation by more than 3 kcal/mol (S1 Data). One example is the sequence GKTLFIS in β-galactase. Paladin predicts the reverse orientation should be favored because it puts all three hydrophobic residues at sites -2, -1 and 0. The five-mer KTLFI has a predicted score of -10.5 kcal/mol, which is more than 3 kcal/mol lower than any other five-mer in either orientation in that sequence. We predict that there are many interesting candidates for further studying the reverse orientation among these peptides.

## Conclusions

Analysis of existing bound structures reveals that substrates bind to DnaK in a highly conserved configuration and that binding specificity likely arises from interactions between side chains on the substrate and distinct regions of the βSBD regardless of the orientation of bound peptide. Motivated by these observations, we derive a new <u>P</u>hysics-b<u>a</u>sed mode<u>l</u> of Dna<u>K</u>-Substrate Bin<u>d</u>ing (Paladin) by calculating a set of energy terms using molecular dynamics simulations to sample the binding configuration of each amino acid bound to each site on the βSBD. These terms were trained to reproduce peptide array data, and the resulting model is able to discriminate binding from non-binding sequences. By avoiding over-training the model on data without information on orientation and limited information about specific binding registry, Paladin is able to accurately recapitulate the specific binding registry and importantly can predict the binding orientation in the limited structures available. Paladin is the first model to consider peptide binding orientation in DnaK, and can discriminate between peptides bound in both forward (N- to C-) and reverse (C- to N-) orientation with ~70% accuracy on the limited data available.

The methodology used to derive Paladin should be more transferable than similar models of Hsp70 binding, due to its physics-based energy terms and minimal training on low-

resolution peptide array data. Paladin's ability to predict substrate binding orientation without any training is evidence of this kind of transferability. We anticipate that the set of weights we selected could perform equally well in a wide variety of slightly different systems, since the model's results does not depend sensitively on the choice of weights, and we carefully avoided overfitting to noisy data. Critically, this prevents the need for additional experimental data that is not available for every system, since one would only need to calculate the energetic terms. This flexibility makes our methodology well-suited for building up models for homologs of Hsp70. An array of predictors mapping the preferences of various homologs of Hsp70s could help tease out the physical rationale for the design of individual Hsp70s and the evolutionary relationships between them as well as to design "super-binder" sequences with enhanced affinity for specific homologs. Furthermore, the possibility of binding of D-amino acids could also be considered. This could become important in design of novel binders, for example, as inhibitors, even though existing data suggest that D-amino acids can bind to DnaJ, a co-chaperone of DnaK in *e. coli*, but not DnaK itself [38,39].

## Methods

### Structural analysis of the binding configuration

When determining the degree of conserved structure, RMSD was calculated after aligning all available structures on the βSBD (residue IDs 389 to 503) using CHARMM as an analysis tool [40]. To compute the RMSD between residues of the forward orientation substrates, the residues corresponding to the same site were compared for each substrate peptide indicated in PDB IDs: 1DKZ, 1DKY, 4EZW, and 4EZX (S1 Fig). Substrates without P and with at least 5 residues were used for this calculation.

### Atomistic simulations and analysis

All simulations and analysis were performed using CHARMM [40–42]. In each simulation, the substrate contains a single residue in a selected initial rotamer state and the rest glycines. DnaK's α-helical lid was considered too far from the binding sites to interact and removed to speed up the simulations, but R467 was restrained to mimic the effect of a salt bridge with lid residue D520 (S15). The backbone conformation is taken from 1DKZ [10] and weakly restrained using harmonic position restraint potentials with force constant of 0.6 kcal mol$^{-1}$ Å$^{-2}$. The force constants were chosen to allow similar backbone fluctuations as observed in the available structures (S1 Table and S1B Fig). The Charmm22 force field was used to describe the system [43]. For computational efficiency and derivation of well-converged energetic quantities, the solvent effects were described using the distance-dependent dielectrics (RDIE) model with dielectric constant of 4. It has been shown that such electrostatic treatment is comparable to more sophisticated continuum electrostatic methods including the full Poisson-Boltzmann equation [44–46] or the one of the Generalized Born approximations [47–49] in protein-ligand docking applications [50]. Such models also cannot be decomposed into pairwise interactions, which is required for deriving a PSSM. Imbalances between in the electrostatic term and other energy terms for charged residues, particularly R, may be related to the use of the RDIE implicit solvent model. Prior to the production simulations, each initial structure was minimized and then equilibrated for 100 ps, during which the temperature was raised gradually from 100 to 300 K. Then, five production simulations of 3 ns were performed to sample interactions and calculate average energies. We can afford to use such short simulations given that each one only samples a different rotamer state, and the backbone is restrained near the pose in the structures (Fig 2A and S1). SHAKE (41) was imposed to constrain the length of all hydrogen containing bonds to enable the use of a 2 fs time step [51]. Analyses

show that the interaction energies for each initial condition converged quickly (S16 and S17 Figs). The final averaged properties were calculated from the last 1.8 ns of the trajectories. We note that sampling of side-chain rotamer states can be limited for large side chains in restricted pockets (e.g., site 0). For this, multiple starting conformations were generated using the backbone-dependent rotamer library published by Shapovalov and Dunbrack Jr [36]. The top-five rotamers in the β-strand basin, (phi, psi) = (-140, 130), were selected for each amino acid. The trajectory with the lowest average total potential energy was selected. All molecular representations were drawn with VMD [52].

## Parameterization of the Paladin model

The Paladin model considers 6 physics-based terms: vdW interactions, electrostatic interactions, backbone strain, desolvation free energy of the substrate side chain and site on DnaK, respectively, and conformational propensity terms for the substrate side chain and backbone. vdW and electrostatic interaction energies between one substrate side chain and the rest of the complex were calculated in CHARMM and directly used as parameters. The total harmonic restraint energy was also used directly to estimate backbone strain in the binding pose. The restraint energy of the complex with an all-glycine substrate was subtracted from each site, giving free-energy cost of moving the backbone to accommodate the side chain. When calculating the constraint energy for P, the restraint on P itself was released to avoid an unrealistically large energetic penalty. The desolvation free energy is estimated from the fraction of total side-chain surface area buried by hydrophobic residues: $E_{desolv} = \frac{-\Delta SASA_{bind}}{SASA_{tot}} \Delta G_{chx \to wat}$, where $\Delta G_{chx \to wat}$ is the experimental free energy of transfer from hexane to water (25). A problem with this approach is that like all other residues, R and K are considered as a whole sidechain rather than being split into their hydrophobic chains and polar head groups. While the single transfer free energy incorporates both of these parts of the side chain, the surface area is not split up according to whether it is on the hydrophobic chain or the polar head group. This means that if the apolar chain is buried, the desolvation penalty is much more unfavorable than it should be. Another important drawback of the SASA model that it fails to describe long-range polar solvents, which won't affect the more solvent-shielded regions but would have increasing effects at the outer sites, particularly on charged residues [35]. The desolvation free energy of the site is the complimentary SASA change for all the neighboring βSBD atoms. We approximated the transfer free energy of backbone atoms using asparagine's transfer free energy. The side-chain conformational propensity was calculated with $E_r = -kT \ln(P_r/P_0)$, where $P_r$ is the probability of the lowest energy rotamer, and $P_0$ is the highest probability. The backbone conformational propensity term $E_{cp}$ is the mean of 1—helical propensity (coil propensity) of the five amino acids and was used to estimate the free-energy cost of confining the substrate backbone conformation to the conformation of the bound state. All of these terms are shifted with respect to $G_5$, so they represent a binding free energy in reference to an all-G backbone.

Weights were determined by a combination of Monte Carlo search to optimize the separation of scores of binders and non-binders. The parameter we optimized was a linear combination of several z-scores: $Z = 4z_{sb-nb} + 2z_{bi-nb} + z_{bi-sb} + 1/2 z_{nu-nb}$, where each z-score separates two classes within the data: $z = \frac{\mu_1 - \mu_2}{\sqrt{\frac{\sigma_1^2 + \sigma_2^2}{2}}}$, where each μ and σ are the mean and standard deviation for a different group in the array data: strong binder (sb), binder (bi), neutral (nu), or non-binder (nb). This dataset is comprised of the full sequences of 37 proteins. It contains 3477 13-mers, of which 7% are strong binders, 23% are binders, 21% are neutral and 48% are non-binders. These classifications were performed by Rüdiger, *et al.* [15], and this data was made available to us by request. A total of ~$10^7$ steps of the Monte Carlo were computed,

where at each step a weight was randomly changed. If the new set of weights improved the Z-score, they were accepted. If not, a random number was drawn and compared to a threshold. If the random number was below the threshold, the new weights were still accepted. Throughout each parallel search, this threshold was incrementally lowered from 0.25 to 0. Ultimately, the site weights were determined to contribute very little to the overall performance, so a set of weights was selected to match the expected relative importance of each site to binding (S7C Fig). All plots were made with python's matplotlib [53]. Paladin is written in python3 and freely available at https://github.com/enordquist/paladin. The script takes an input file that contains one or more FASTA-format sequences. Every five-mer in the input sequence(s) will be scored. The computational cost of Paladin is minimal. For example, the Rüdiger dataset can be scored in 0.8 seconds on a single Intel Xeon E5-2620 CPU at 2.10GHz.

## Comparison with other predictors

Where relevant, the same predictor evaluations (ROC, Precision-Recall, register and orientation prediction) were performed using each previous algorithm (Rüdiger, Limbo, BiPPred and ChaperISM). In all the tests, every (model-specific) minimally-sized peptide is scored for each protein sequence. Rüdiger, *et al.* model was not originally designed to score short (about seven to ten) peptides with respect to site-specific register or orientation, so it wasn't initially evaluated on either. Similarly, ChaperISM shouldn't by default be used to consider register or orientation, although ChaperISM was evaluated on register prediction alongside Limbo as a benchmark. The datasets of each register prediction are limited to peptides with the minimal number required by each algorithm. Specifically, Paladin and BiPPred each score five-mers and require a minimum of five residues for an orientation prediction and six for a register prediction. Limbo similarly requires seven and eight for orientation and register predictions, respectively. Limbo scores were obtained from the free, online servers at https://limbo.switchlab.org/ using the "Best overall prediction" setting and default binder cutoff at 15. BiPPred scores were obtained from the free, online server at https://www.bioinformatics.wzw.tum.de/bippred/submit/ using default settings. ChaperISM is freely available from github (https://github.com/BioinfLab/ChaperISM). Scores were generated in qualitative mode (better for classifications than quantitative mode, per publication) with a default cutoff at 0.2.

## Supporting information

**S1 Table. Available structures of DnaK βSBD-substrate complexes.** The amino acid residue bound at site 0 is bold. Forward means the peptide is bound N- to C- as in Fig 2. Z stands for cyclohexlalanine.
(XLSX)

**S2 Table. Paladin scores for substrates in of DnaK SBD-substrate complexes.**
(XLSX)

**S3 Table. Paladin 5x20 PSSM.** These scores are derived by multiplying the energy terms in Figs 4 and S4 by the weights in Table 1. These have the unit kcal/mol. The PSSM is displayed visually in S5 Fig.
(XLSX)

**S4 Table. Area under the Precision-Recall curves (given in S9 Fig).**
(XLSX)

**S1 Movie. Conformations and backbone hydrogen boning interactions of substrate in forward and backward orientations.**
(MP4)

**S2 Movie. Substrate binding surface around site 0.**
(MP4)

**S1 Data. Top 100 reverse orientation predictions from Rüdiger, et al. dataset.**
(XLSX)

**S1 Fig. Backbone RMSD of available structures.** (A) RMSD of βSBD backbone atoms. (B) RMSD of the backbone of residues of substrates without prolines by site (1DKY, 4EZW, 4EZX). For both figures, the reference structure is 1DKZ. All structures were aligned to 1DKZ based on residues 393–503 (βSBD).
(TIF)

**S2 Fig. Site 0 Sidechain Snorkeling.** The two βSBD residues E402 and T437 have solvent exposed carbonyls which can potentially satisfy and stabilize charged and polar residues buried in site 0, illustrated here with L. This is a snapshot from an MD simulation.
(TIF)

**S3 Fig. Bridging M404 with A429.** (A) The interaction between the bridging M404 and A429, which together partially cover site 0, viewed from the site -1 of the complex. (B) The same as (A) but viewed from the site +1 side. The structure shown is PDB ID: 1DKZ.
(TIF)

**S4 Fig. MD-derived interaction parameters by term for each site.** (A) site +2, (B) site +1, (C) site -1, and (D) site -2. Site 0 is included in the main text (Fig 4).
(TIF)

**S5 Fig. Paladin 5x20 PSSM.** These scores are derived by multiplying the energy terms in Figs 4 and S4 by the weights in Table 1. The values from this figure are also given in S3 Table.
(TIF)

**S6 Fig. Monte Carlo optimization of term weights.** Top 1000 Z-scores vs various weights from all Monte Carlo runs (see Methods). The weights correspond to (A) vdW interaction energy, (B) Electrostatic interaction energy, (C) Backbone strain, (D) Desolvation FE of the sidechain, (E) Backbone conformational propensity, and (F) Sidechain conformational propensity.
(TIF)

**S7 Fig. The Site and Term Weights are interdependent.** The plots show two weights as a function of Z-score along the color axis for the same top 1000 results from all Monte Carlo runs and additional grid searches as discussed above. The panels show the weight combinations: (A) Desolvation for the substrate side chain and van der Waals, (B) Desolvation of the substrate side chain and electrostatics, and (C) Site 0 and site -2, as functions of the Z-score (see Methods).
(TIF)

**S8 Fig. Example peptide array of Luciferase and predictions of Rüdiger, *et al.* and Paladin models.** (A) The raw peptide array data from Rudiger, *et al.* [15]. (B) Predicted peptide array from Rudiger, *et al.* model. Scores were min-max normalized with thresholds of -3 and 6 in order to clarify the distribution. (C) Predicted peptide array from Paladin model. Min scores for each 13-mer were min-max normalized with thresholds of -10 and -6 to clarify the

distribution. Refer to the Methods section in the main text or original publication [15] for details about peptide arrays.
(TIF)

**S9 Fig. Precision-Recall Curves for peptide array predictions.** The plots are divided in the same manner as Fig 5 in the main text, with (A) comparing all binders and non-binders and (B) comparing only strong binders and non-binders. The solid lines represent the training set, the solid lines represent test set. The dashed red line at is the fraction of true positives (all binders or strong binders) in the data and reflects the precision of a random predictor. The corresponding areas under the curves (AUC) are given in S4 Table.
(TIF)

**S10 Fig. Binding register prediction comparison for Limbo and ChaperISM.** Forward-orientation register predictions for (A) Limbo [16] and (B) ChaperISM [21], which both score 7-mers in only the forward (N- to C-). As in Fig 6 in the main text, stars denote when a 7-mer is "correct", that is when it is the 7-mer bound in the crystal structure, whereas circles denote all other 7-mers. Both of these algorithms score 7-mers that bind tightly with a higher score (opposite from Paladin's energy-like score).
(TIF)

**S11 Fig. Correlation of Substrate side-chain interaction energy terms at site 0 in the context of forward and reverse backbone conformation.** To calculate the vdW, electrostatic and backbone conformational propensity energy terms at site 0, we performed the identical simulation procedure outlined in Methods section of the main text, except we used the backbone position in 4EZY, (NRLILTG bound C- to N-). We simulated the side chains of: A, L, I, Y, W, N, E, K, and R. Error bars are calculated as standard error of the mean.
(TIF)

**S12 Fig. Limbo Binding Orientation Prediction.** Negative delta (forward–reverse scores) indicates that the reverse orientation is predicted to be preferred, and positive delta for the forward orientation. LIMBO scores peptides of length 7 and uses a higher number to indicate higher affinity to DnaK [13].
(TIF)

**S13 Fig. BiPPred Binding Orientation Prediction.** Negative delta (forward–reverse scores) indicates that the reverse orientation is predicted to be preferred, and positive delta for the forward orientation. BiPPred scores peptides 0 to 1, where a higher number indicates higher affinity to the endoplasmic Hsp70 BiP [26].
(TIF)

**S14 Fig. Reverse orientation register predictions.** As in Fig 6 in the main text, the Paladin scores are shown for each substrate peptide and all possible binding registries that allow all five sites on DnaK occupied. Both backbone orientations are considered and denoted by color (forward: blue; reverse: orange). The registries as observed in the PDB structures are shown with orange stars.
(TIF)

**S15 Fig. Effect of deleted helical lid.** (A) R467 without a salt-bridge (D520-R467) is easily able to interact with E at site -1 of the substrate, resulting in considerable electrostatic contributions that were unexpected. (B) Restrained to mimic effect of salt bridge with lid. Interactions are more as expected (and as reported in parameters for site -1).
(TIF)

**S16 Fig. Convergence of Interaction Energies.** To demonstrate that the length of simulations is appropriate, here is an example of traces for a simulation run for L at site 0 that was 50 ns long, roughly 10 times the length we thought would be necessary. The interactions plotted are (A) Total Interaction Potential energy (kcal/mol), (B) Van der Waals, (C) Electrostatic, (D) Harmonic restraint, and (E) Solvent Accessible Surface Area ($\mathring{A}^2$). The slight transition just before 30 ns corresponded to a conformational change from the rotamer basin corresponding to the initial configuration into a secondary conformational basin. In fact, all of the short simulations of leucine (5 total, started from 5 rotamers) ended up in similar regions to these to rotamer basins.
(TIF)

**S17 Fig. Convergence of Total Energies.** The lowest potential energy traces at site 0 for (A) L and (B) R, respectively. The average total energy was calculated for the last 3/5 of the 3 ns dynamics for each of rotamer-initiated simulation. The lowest-energy trajectory was used to calculate the interaction energies for each site.
(TIF)

# Acknowledgments

The authors thank Dr. Stefan Rüdiger for kindly providing the peptide array experimental data used for model training.

# Author Contributions

**Conceptualization:** Woody Sherman, Lila M. Gierasch, Jianhan Chen.

**Formal analysis:** Erik B. Nordquist, Jianhan Chen.

**Investigation:** Erik B. Nordquist, Charles A. English.

**Methodology:** Erik B. Nordquist, Charles A. English, Woody Sherman, Jianhan Chen.

**Writing – original draft:** Erik B. Nordquist, Jianhan Chen.

**Writing – review & editing:** Erik B. Nordquist, Charles A. English, Eugenia M. Clerico, Woody Sherman, Lila M. Gierasch, Jianhan Chen.

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
