## [Decision Letter · Decision Letter 0]

22 Jun 2021

Dear Dr. Chen,

Thank you very much for submitting your manuscript "Physics-Based Modeling Provides Predictive Understanding of Selectively Promiscuous Substrate Binding by Hsp70 Chaperones" for consideration at PLOS Computational Biology.

As with all papers reviewed by the journal, your manuscript was reviewed by members of the editorial board and by three independent reviewers. In light of the reviews (below this email), we would like to invite the resubmission of a significantly-revised version that takes into account the reviewers' comments.

We cannot make any decision about publication until we have seen the revised manuscript and your response to the reviewers' comments. Your revised manuscript is also likely to be sent to reviewers for further evaluation.

Sincerely,

Guanghong Wei

Associate Editor

PLOS Computational Biology

Nir Ben-Tal

Deputy Editor

PLOS Computational Biology

Reviewer's Responses to Questions

**Comments to the Authors:**

Reviewer #1: Summary:

Here, Nordquist et al. are interested in develop a Hsp70 peptide binding prediction method based on physics-based simulations to identify client proteins that can bind to SBD region. In their method, they were also able to identify the orientation of the peptides inside the cleft (N-C – “forward”; C-N - “reverse”). The authors also want to provide molecular details of the client protein-SBD binding. The research question is sound and relevant to the field.

Current predictors failed to show molecular details on positions occupied in the SBD or the orientation of the bound peptide. Here, the authors try to fill this gap using their approach. In terms of predicting binding, Paladin performs similar (but still worst according to AUC ROC curve values) to other methods. One positive effect from its training models, possible through transfer learning mechanisms, was the ability to predict the orientation of the peptides (forward vs reverse) in the SBD cleft with accuracy around 70%.

In my opinion, Paladin come to contribute with the field as a new Hsp70 binding predictor, and the main novelty is its capacity to differentiate to some extent the orientation of the peptides, something that is not performed by current predictors.

In light of these facts and the comments below, I recommend the acceptance of this paper with minor revisions.

Abstract and introduction:

Strengths:

- The authors summarize the main research question and key findings.

- The authors covered current literature on the field and explained how previous findings relate to their study.

Weaknesses:

- None

Results and conclusions:

Strengths:

- The results support the conclusions.

- Limitations are discussed.

Weaknesses:

- Figure 2B is not sufficient to support this statement on Page 8, line 145-147. Here, I suggest to compare these two orientations with a different approach (e.g. peptide residue-SBD distance measures).

- In section “Binding site geometry and surface properties determine the DnaK-substrate selectivit”, the authors mention predictions from LIMBO and Rudiger, but not ChaperISM.

- On Page 11, line 233, there is no graph or values showing correlation between vdW interaction energies and size of the side chain, only visual inspection of Figure 4 and S4.

- On Page 12, line 235-237, the authors should comment on the possible impact of the length of the simulation (3ns) on the backbone strain term.

- On Page 15, line 303-305, authors should give details on the data set balance. If the data set is imbalanced, it may be more interesting to show a precision-recall curve. In case the data set is balanced, it would be interesting to show how Paladin would perform in a imbalanced idependent data set (look CD data set at doi:10.1002/prot.25084). Despite this data set came from a different protein, the authors claim that their method can be expanded to other Hsp70 chaperones.

- The AUC ROC values for Paladin, in comparison with Rudiger and ChaperISM, performs worst. The authors have provided possible reasons why this is happening, saying “...the small differences in ROC curves of Paladin and previous models are likely insignificant.”. However, none of these reasons were tested.

- On Page 20, line 417-418, the authors should also mention how ChaperISM performs with Pro at site 0.

- The final PSSM matrices with energetic terms need to be included, as this is common practice for prediction methods.

Figures and tables:

It is of my opinion that the figures should be carefully revised, either in terms of resolution and labels, but specially in terms of legends. I will give some examples here:

- Figure 1: Since ATP and ADP play an important role and it is mentioned in the text and in the legend of the figure, these molecules should be shown in the figure.

- Figure 2: The interpretation of image B is difficult. I'd recommend (i) different colors for each one of the orientation modes or (ii) two separate images, one with the forward orientation and another one with the reverse orientation.

- Figure 3: There is room for improvement in this image, specially in respect to the surfaces and the representation of the peptides (e.g. ribbon vs stick; colors; transparency).

- Figure 4: I suggest to increase the space in between the charts. Also, the legend is interfering with the data in some cases (e.g. Backbone Conformational Propensity). I suggest to move the legend to another place. Additionally, what FE stands for? This should be written in the legend.

- Figure S2: Label the residues in the figure. This is a important figure, however, the representation and labels are not adequate to show the relevant point.

- Figure S3: Label the residues in the figure.

- Figure S4: The graph for "Backbone Conformational Propensity" is the same in A-D. Is this correct? If yes, why is so different from site 0?

- Figure S7: Correct axis labels.

- Figure S8: The legend of the figure do not explain what the star and the spheres represent. “Correct” and “other” may not be the better label, unless explained in the legend.

- Figure S12: Label the residues in the figure.

- Figure S13: Please, correct y-axis titles appropriately (e.g. type of energy and unit).

- Figure S14: Correct Y-axis (missing parenthesis).

- Table 2: table caption is incomplete.

- Table S1: Include a footnote to explain "Z" amino acid corresponds to cyclohexyalanine.

Methods:

Strengths:

- The code and additional files are freely available through GitHub.

- Information about the atomistic simulations are complete.

- Parameterization of Paladin model is explained thoroughly.

Weaknesses:

- The authors should include:

- a section showing how the structure conservancy was assessed (this is in respect to the first result).

- one section explaining how the comparison among methods (limbo, rudiger, chaperism, etc) was performed. Also, explain the distribution of the data on the used datasets (e.g. balanced vs imbalanced).

- README file on github should be updated with more information, specially about the output scores.

Minor Issues:

- Introduction is well-written. Results, however, should pass through some language editing to improve clarity.

- Standardize the nomenclature of amino acids (i.e. one letter vs three letter code).

- Page 7, line 112: Please check if reference 25 is the correct one for this sentence.

- Page 8, line 154: Please, correct the “Figure 3” statement.

- Page 12, line 249: Misplaced reference (ChaperISM reference).

- Page 16, line 332: “Figure 5” should be replaced by “Figure 6”.

- Page 16-17, lines 338-341: Repeated text about ChaperISM.

- Page 19, line 395: In Figure 7 I counted 18/22, and not 16/22. Which one is the correct?

Reviewer #2: Summary

In this work, the authors develop a physics-based algorithm to predict binding sites of DnaK, the Hsp70 from E. coli. They posit their algorithm is unique in comparison to existing tools in that it is able to predict orientation (Forward versus reverse) as well as registry (i.e. the specific site occupied by each substrate residue in the DnaK substrate-binding-domain).

In their first set of results, the authors perform a qualitative analysis of a set of previously determined DnaK structures by Zahn and colleagues, and discuss which residues are represented in each position in the substrate and what the underlying energetics are. They also cross-reference these data with LIMBO, a comparable DnaK-substrate prediction algorithm.

Next, the authors describe and validate their own prediction algorithm, Paladin. They show that their algorithm performs similarly to existing algorithms in terms of predicting DnaK-binding peptides in a dataset generated by Rüdiger et al. They further show that Paladin can predict registry of forward-binding peptides, and can predict whether a peptide will bind in the forward or the reverse orientation based on a relatively small set of DnaK-substrate structures.

The authors conclude that the algorithm they have designed is a good DnaK substrate predictor, and given the relatively small extent to which the algorithm was parameterized on the experimental data set, they predict the algorithm to be easily transferable to similar systems.

General

The physics-based algorithm described here adequately predicts DnaK-binding sites, and performs comparably to existing tools for the same purpose. Their use of a 5-residue binding site allows the authors to test for specific substrate registry, in which Paladin outperforms some of its competitors. The algorithm also predicts binding orientation reasonably well, although the authors do not compare the performance therein with other predictors. The explicit use of individual force terms allows for an assessment of which forces drive binding of certain peptides, which is not the case for predictors such as the one devised by Rüdiger, but is comparable to LIMBO. Another advantage of the algorithm seems to be its transferability to similar systems, although the authors should test this before claiming it. Furthermore, the authors provide an interesting analysis of how and why DnaK can interact with substrates in two opposite orientations.

Overall, the authors describe a tool that satisfactorily predicts DnaK binding sites, their binding orientation, and in specific cases, their registry. Although it is not entirely novel and does not massively outperform competitors, the authors describe an additional approach to tackling the important question of DnaK-substrate interactions, and yield additional insight into binding orientation and registry.

Major remarks

Line 132: in order to determine how conserved the SBD conformation is, the authors align the structures from Table S1 and determine the RMSD to be small. However, they mention that they only look at structures with substrates in the forward orientation AND structures in which the substate does not contain Pro. Does this mean they have omitted 13 structures out of the 18 forward-oriented structures? In that case, this analysis hardly shows the SBD structure to be highly conserved. Also, does this mean the authors find the SBD needs substantial conformational changes in order to accommodate Pro? This would be interesting information and the authors should comment on this.

Line 146: The authors mention that the SBD conformation is highly conserved between forward- and reverse-oriented substrates, though they never test for this since they don’t use reverse-oriented substrates in their RMSD analysis. It would be useful to include such an analysis to show that not only the substrate sidechains are located in roughly the same positions, but also the SBD remains in largely the same conformation.

In their construction of a physics-based model, the authors consider 6 physical forces, but don’t mention hydrogen bond formation at all. Could the authors comment on why they omit a crucial force in protein-protein interaction from their model, and why they feel this choice is justified?

Line 242-249: How do the authors reconcile the fact that they see only a minor preference for Lysine and no preference for Arg whatsoever, although preferences for both basic references have been described in literature, including by Rudiger et al, and that Arg is represented in many DnaK-substrate interaction structures?

The authors compare Paladin to LIMBO for registry prediction, and find that LIMBO performs similarly to Paladin. However, they never test whether LIMBO is also capable of testing for orientation, and simply show results for their own algorithm. It would be interesting to see if Paladin is uniquely capable of registry prediction, or whether other tools can also do this.

The authors claim the minimal training would allow Paladin to be easily transferable to similar systems. To show this, it should be straightforward to use the experimental data used for the production of the BiPPred tool, and test whether a paladin-type PSSM would also perform well in this system (without retraining on the experimental data, and only using structural information and the weights determined for this algo).

Minor remarks

Abstract: “the chaperone is specific amino acids can vary considerably”, this phrase doesn’t make sense.

Line 48: “Paladin provides a physical basis to understand why and how DnaK binds specific peptides”. The how is clear, I.e. the molecular modelling gives some information in the driving forces of binding, although some, like H-bonding, are ignored. The “Why” might be a stretch, since the authors don’t directly link the hsp70 binding preferences they observe to substrate protein characteristics that warrant hsp70 interaction.

Line 67: “slow on/off binding” should be changed to “slow on/off” rates

Figure 2 caption: “hydrogen bonds are shown in magenta with matching size to their corresponding substrate”. This is a little unclear, I assume the sizes of the hydrogen bond lines are thick/thin depending on whether they are formed with forward/reverse orientation. The differences are hard to make out in the figure

Line 154: “Figure 3” should be in brackets

Line 175: should the “+1” here be “-1”?

Lines 188-199. It would be interesting if the authors address whether charge-charge interactions stabilize the Args that are observed in structures at positions -2 and +2. And if they don’t, which interactions do stabilize them? Given the general tendency for hsp70s to bind basic residues, this would be interesting to elaborate upon.

Line 225: does “the site” refer here to the DnaK binding site? This is unclear.

Line 459: R4467 should be R467

Line 495-496: “A problem with this approach is that some residue side chains, such as those of Arg and Lys, which are hydrophobic chains with polar head groups.” This is not a full sentence. Also, the authors state this is a problem, but don’t offer a solution for it. Would this offer an explanation as to why Arg is not favored in their algorithm, yet has been found in binders in vitro, as well as in DnaK-substrate structures?

Figure S7C: there seems to be a yellow square in the top right hand corner of each of the plots, which it seems is not supposed to be there?

Lines 338-340: The authors repeat themselves here, one of these sentences should be deleted.

Line 429: “revealS”

It would be useful for the authors to describe how Paladin could be used on sequences of entire proteins. In this work, they design the algorithm and train it on existing experimental data. However, the power of such predictors lies in determining putative binding sites that have not been experimentally validated. Is Paladin capable of screening entire proteins and even proteomes for putative binding sites, and how (e.g. a sliding window approach)? And can it be used to that end by users in its current form?

Reviewer #3: Nordquist et al provide a predictor for binding sites of the bacterial Hsp70 chaperone, DnaK, based on molecular dynamics simulations. Hsp70 chaperones are key for the cellular protein folding machinery. The bacterial DnaK homologue is the best understood paradigm for the overall mechanism of action of this family. Key to understand its function in protein folding is to understand how it recognises its substrates. Prediction of such sites is possible since 1997, based on empirical analysis of peptide sequences binding to DnaK.

The present study has a different approach, as it bases the prediction on the analysis of the substrate binding site of DnaK. This conserved Hsp70 binding sites comprises binding positions for five consecutive residues. Contacts are made by both the side chains and the backbone. Additionally, coulomb interactions outside the binding pocket contribute to affinity. The authors test their model against the original data base from the 1997 study.

Their prediction is slightly less accurate than the 1997 algorithm. However, there are two main selling points: (I) The predictor starts from analysing the chemical properties of the chaperone binding site, not the substrate. This may result in more insights into how substrate recognition of Hsp70 works. (II) The predictor can predict the orientation of the peptide stretch in the Hsp70 binding pocket. Thus, if sufficiently accurate this new predictor is an interesting new tool to understand Hsp70 substrate interactions. There are some concerns the authors need to address before publication.

Concerns and advice

1. The predictor does not make full use of the information in the available experimental material.

a. It neglects the flanking regions outside the 5-residue binding site. These regions can contribute via coulomb interactions, and considering their contribution improves the prediction in the Rüdiger algorithm. Thus, taking this into account would improve the quality of prediction.

b. The manuscript states that there is not information in the peptide data about where in a peptide DnaK may bind. This information can in fact be extracted from the peptides binding data by Rüdiger et al, 1997, as overlapping peptides allow to locate the highest affinity segment when comparing neighbouring peptides. It would allow the authors to test whether their algorithm would predict correctly the common core in overlapping peptides. How does it do on this?

2. Some parameters in the prediction matrix do not fit previous experimental data.

a. The negative effect of acidic residues in the 5-residue binding site only has a moderate effect, while positive charges are dramatically disfavoured. This is contrast to both the statistical composition of the binding peptides and the finding that negatively charged residues wipe out binding, while positively charged residues do not. This suggests that the MD analysis of the present study does not adequately takes charges into account.

b. The structural basis of the predictor is based on the Hendrickson structure of the DnaK substrate binding domain. This study describes that the central position 0 is tailored for Leucine. This is consistent with the analysis of the composition of the residues in this region, which shows Leu strongly favoured over all other residues. This is not the case for the predictor here, and it is unclear why. Here Ile is almost as good as Leu.

3. The peptide data to analyse forward/reverse binding are based on a relatively small number of peptides, which also appear relatively untypical. The impact of this is not adequately discussed, nor are data taken into account assessing the impact of D-amino acids on DnaK backbone binding (Rüdiger et al, 2001; Feifel et al, 1998).

4. There are experimental data available on the impact of the hydrophobic arch over the substrate binding cleft on affinity and specificity (Mayer et al, 2000, Rüdiger et al, 2000). These are not taken into account nor adequately discussed.

**Have the authors made all data and (if applicable) computational code underlying the findings in their manuscript fully available?**

Reviewer #1: **No: **The final PSSM matrices with energetic terms need to be included, as this is common practice for prediction methods.

Reviewer #2: Yes

Reviewer #3: Yes

PLOS authors have the option to publish the peer review history of their article (what does this mean?). If published, this will include your full peer review and any attached files.

Reviewer #1: No

Reviewer #2: No

Reviewer #3: **Yes: **Stefan G.D. Rüdiger
---

## [Decision Letter · Decision Letter 1]

20 Sep 2021

Dear Dr. Chen,

Thank you very much for submitting your manuscript "Physics-Based Modeling Provides Predictive Understanding of Selectively Promiscuous Substrate Binding by Hsp70 Chaperones" for consideration at PLOS Computational Biology. As with all papers reviewed by the journal, your manuscript was reviewed by members of the editorial board and by several independent reviewers. The reviewers appreciated the attention to an important topic. Based on the reviews, we are likely to accept this manuscript for publication, providing that you update your point-by-point reply that Reviewer #3 required and this reviewer's previous comments were adequately addressed.

Please submit your updated point-by-point reply and revised manuscript within 30 days. If you anticipate any delay, please let us know the expected resubmission date by replying to this email.

Sincerely,

Guanghong Wei

Associate Editor

PLOS Computational Biology

Nir Ben-Tal

Deputy Editor

PLOS Computational Biology

[LINK]

Reviewer's Responses to Questions

**Comments to the Authors:**

Reviewer #1: My previous recommendation was "Accept with minor revisions". Since the authors answered all my questions and modified the text accordingly, I'm thereby recommending this article for publication.

Reviewer #2: All my comments were adequately addressed.

Reviewer #3: The authors provided an extensive and detailed point-by-point reply. Unfortunately, the page numbers and line numbers do not match, and the positions indicated do not contain the arguments the authors are referring to. I would appreciate if the authors would provide an updated point-by-point reply that would allow me to follow their argumentation.

**Have the authors made all data and (if applicable) computational code underlying the findings in their manuscript fully available?**

Reviewer #1: Yes

Reviewer #2: Yes

Reviewer #3: Yes

PLOS authors have the option to publish the peer review history of their article (what does this mean?). If published, this will include your full peer review and any attached files.

Reviewer #1: No

Reviewer #2: No

Reviewer #3: **Yes: **Stefan G.D Rüdiger

Figure Files:

Data Requirements:

Reproducibility:

References:

---

## [Decision Letter · Decision Letter 2]

15 Oct 2021

Dear Dr. Chen,

We are pleased to inform you that your manuscript 'Physics-Based Modeling Provides Predictive Understanding of Selectively Promiscuous Substrate Binding by Hsp70 Chaperones' has been provisionally accepted for publication in PLOS Computational Biology.

Best regards,

Guanghong Wei

Associate Editor

PLOS Computational Biology

Nir Ben-Tal

Deputy Editor

PLOS Computational Biology

Reviewer's Responses to Questions

**Comments to the Authors:**

Reviewer #3: The authors carefully addressed all reviewers' comments. The implanted changes now provide an adequate overview on how the new algorithm compares to the experimental data. It is interesting to see how structure-based predictions zoom in further on the hsp70 binding motif.

**Have the authors made all data and (if applicable) computational code underlying the findings in their manuscript fully available?**

Reviewer #3: Yes

PLOS authors have the option to publish the peer review history of their article (what does this mean?). If published, this will include your full peer review and any attached files.

Reviewer #3: **Yes: **Stefan Rüdiger

---

## [Editor Report · Acceptance letter]

29 Oct 2021

PCOMPBIOL-D-21-00933R2 

Physics-Based Modeling Provides Predictive Understanding of Selectively Promiscuous Substrate Binding by Hsp70 Chaperones

Dear Dr Chen,

I am pleased to inform you that your manuscript has been formally accepted for publication in PLOS Computational Biology. Your manuscript is now with our production department and you will be notified of the publication date in due course.

With kind regards,

Livia Horvath
